# Chikungunya in Infants and Children: Is Pathogenesis Increasing?

**DOI:** 10.3390/v11030294

**Published:** 2019-03-23

**Authors:** Kelli L. Barr, Vedana Vaidhyanathan

**Affiliations:** 1Department of Biology, Baylor University, Waco, TX 76798, USA; 2Vedana Vaidhyanathan, Central Libraries Research Engagement, Baylor University, Waco, TX 76798, USA; Vedana_Vaidhyanathan@baylor.edu

**Keywords:** Chikungunya virus, congenital infection, perinatal infection, neuroinvasive disease, pathogenesis, cutaneous lesions

## Abstract

Chikungunya virus (CHIKV) was first extensively described in children during outbreaks in India and South Asia during the mid-1960s. Prior to the 2005 emergence of CHIKV on Reunion Island, CHIKV infection was usually described as a dengue-like illness with arthralgia in Africa and febrile hemorrhagic disease in Asia. Soon after the 2005 emergence, severe CNS consequences from vertical and perinatal transmission were described and as CHIKV continued to emerge in new areas over the next 10 years, severe manifestation of infection and sequelae were increasingly reported in infants and neonates. The following review describes the global reemergence and the syndromes of Chikungunya fever (CHIKF) in infants and children. The various manifestations of CHIKF are described and connected to the viral lineage that was documented in the area at the time the disease was described. The data show that certain manifestations of CHIKF occur with specific viral lineages and genetic motifs, which suggests that severe manifestations of CHIKF in the very young may be associated with the emergence of new viral lineages.

## 1. Introduction

Chikungunya (CHIKV) is an alphavirus usually vectored by *Aedes aegypti* and *Aedes albopictus* mosquitoes [1]. CHIKV originated in Tanzania and is most closely related to o’nyong nyong virus, which originated in Uganda [2]. CHIKV is a small, enveloped virus with a single-stranded, positive-sense RNA genome approximately 12 kb long, which is divided into two open reading frames. The first section comprises the four non-structural proteins (nsP1, nsP2, nsP3, and nsP4) that are responsible for viral replication inside the host cell cytoplasm. The second open reading frame encodes the structural proteins (capsid, envelope 3 (E3), envelope 2 (E2), 6K, and envelope 1 (E1)). CHIKV can be classified into three genetic lineages based upon sequences of the E1 gene [3]. 

Since first reported in Tanzania in 1952, CHIKV has been identified in multiple outbreaks as a cause of dengue-like illness with arthralgia in Africa and febrile hemorrhagic disease in Asia [4,5,6,7,8,9,10,11]. Prior to the 2005 emergence of CHIKV on Reunion Island, chikungunya fever (CHIKF) was commonly misdiagnosed as dengue [12,13]. Misdiagnosis was common as both viruses occupy the same ecological niches and exhibit similar disease syndromes. In 2005, congenital and perinatal CHIKV infections were reported following delivery to mothers with documented viremia accompanied with symptoms of CHIKF [14,15,16]. Both congenital and perinatal CHIKV transmissions are directly associated with infected mothers. In this review, congenital transmission is indicated when viremia and symptoms are present in mothers at least one week to several weeks prior to birth and perinatal transmission is designated when mothers exhibit symptoms with viremia within a few days prior to or following delivery. Pediatric infection is indicated in individuals over two years of age. Here, reports of congenital and perinatal CHIKF are discussed as incidence of these infections, has occurred more frequently and have increased in severity since the 2014 global expansion (Figure 1). There is a significant need to understand CHIKV pathogenesis in children and neonates because the newly emergent genotypes of CHIKV cause multiple severe manifestations of infection, which can have significant, permanent consequences (Figure 1). Whether this is due to a change in the virus or just better diagnostics and reporting has not been proved.

The following review describes the global reemergence and the syndromes of CHIKF in neonates, infants, and children. The various manifestations of CHIKF in infants and children are described and matched to the viral lineage that was documented in the area at the time the disease was described. The data show that certain manifestations of CHIKF occur with specific viral lineages and genetic motifs, which suggest that severe manifestations of CHIKF in the very young may be associated with the emergence of new viral lineages.

## 2. CHIKV in Neonates and Children during the 1900s

CHIKF was first extensively described in children during outbreaks in India and South Asia during the mid-1960s (Table 1) [17,18,19,20,21]. In Thailand, CHIKF was described as a dengue-like illness with fever, rash, arthralgia, and myalgia, with occasional hemorrhagic manifestations [5,20,21,22,23,24]. In a cohort of 148 CHIKV-exposed children, Vu Qui et al. [18] described multiple dengue-like illnesses in the Saigon region that were difficult to diagnose during the acute period due to similar disease symptoms with dengue and that viral neutralization tests were necessary for a definitive diagnosis. During the 1964 CHIKV outbreak in Madras, India, Jadhav et al. [17] describe pediatric CHIKV infection in great detail. For 33 infants with a dengue-like illness, 11 were admitted to the hospital. All 11 infants had fevers with five having temperatures greater than 104 °F [17]. Ten of the 11 infants developed a macular erythematous rash [17]. Diarrhea was seen in four infants and severe arthralgia in one infant [17]. Two infants experienced febrile seizures [17]. Later studies explained that the outbreaks in Thailand and Vietnam were caused by CHIKV of the Asian lineage [25].

During the 1970s, additional clinical features were described for children and infants with CHIKF (Table 1). The 1974 CHIKV epidemic in Ibadan, Nigeria was caused by the West African genotype and was described as a febrile illness with rash and joint pain from which all children recovered [26,27]. During 1970–1972, CHIKV of Asian lineage caused hemorrhagic fever epidemics in children in Burma. The symptoms reported were indistinguishable from dengue and most infected individuals also presented with similar pulmonary hemorrhagic symptoms as Influenza A and B, which were also circulating at the time [9,10,25,28].

For the most part, prior to the outbreak in Reunion Island in 2005, CHIKF in children and the very young typically involved biphasic fever and rash with the occasional description of pain and/or hemorrhage (Table 1) [17,18,19,26,27]. Although neonatal and pediatric seizures and other CNS symptoms were reported in children during the early outbreaks, the onset of such symptoms was invariably linked with fevers greater than 104 °F [5,17,19,26,29].

It is of interest that arthralgia was less frequently reported for CHIKF caused by the Asian lineage than the African lineages (Table 1). Furthermore, hemorrhagic fever was more commonly diagnosed in children infected with CHIKV of the Asian lineage (Table 1). While genetic motifs related to viral pathogenesis haven’t been described for these three lineages, a unique V386G mutation was identified in the E2 gene of the Asian phylogroup [3]. The change from highly non-polar valine to a small polar glycine could have significant effects on protein structure and function though, since this particular residue is buried next to the capsid and is not likely to play a part in pathogenesis [30]. It is likely that there are other lineage-specific motifs that correlate with disease.

## 3. CHIKV in Neonates and Children after Re-Emergence

Soon after the emergence of CHIKV on Reunion Island, congenital transmission of CHIKV was hypothesized following reports of CHIKF in infants 2–4 days old because viremia had been shown to occur between 3–5 days following infection [98,99]. Beginning in 2006, congenital and perinatal transmission of CHIKV was reported in Reunion Island (Table 1) [14,15,16,98]. These reports were soon followed by thorough descriptions of CHIKF in patients from a few days old up through 18 years of age [36,100]. Here, most infants less than six months of age exhibited rash and dehydration while children between 3 and 18 years old did not exhibit rash, but, instead, were significantly more likely to have neurologic and digestive symptoms [36,101].

Severe CNS consequences from vertical and perinatal transmission were described for the 2005 Reunion Island outbreak and the 2006 outbreak in Bellary, India [16,36,37,102,103]. CNS involvement such as complicated seizures and abnormal MRI findings were reported in over half of patients [37,102]. Cardiac defects were identified in almost half of patients and included myocardial hypertrophy, ventricular dysfunction, pericarditis, and coronary artery dilatation [102]. In infants, these conditions can lead to death in the first year of life without significant medical intervention [104,105]. Touret et al. [15] reported three congenital infections that occurred between 12 and 15 weeks of pregnancy during the 2005 Reunion Island outbreak [15]. While all fetuses died, there were no apparent malformations. Lenglet et al. [14] also showed that congenital CHIKV infection could result in miscarriage if it occurred within the first 22 weeks of pregnancy. While they showed that, after 22 weeks, congenital infection did not cause any apparent defects, there was about 50% perinatal infection risk for mothers with viremia at delivery though he did not report significant consequences or sequelae in offspring [14].

At that time, it was shown that the Reunion Island outbreak strain possessed an alanine to valine substitution at residue 266 in the E1 protein [3]. It was also of note that all patients with symptoms of neuroinvasive disease were infected with this A226V variant strain (21). The A226V mutation has been shown to enhance the transmission of CHIKV in *Ae. albopictus* mosquitoes which has significantly increased the geography of the virus [106]. Viral pathogenesis studies of this variant showed that CHIKV localized to neuronal cells of the cerebellum and induced a significant upregulation of Toll-like receptor-3 (TLR3) in mice [107]. However, these studies employed a strain with several other mutations such that the precise role of the A226V mutation in CHIKF cannot be defined.

During the 2006–2007 CHIKV outbreak in Sri Lanka, children born to women infected with CHIKV during the first two trimesters of pregnancy had less than 50% chance of a healthy birth [38]. Many neonates infected during the first and second trimester exhibited abnormalities including hyperpigmentation, pre-term birth, fever, and various cardiac defects including atrial septal defect, patent ductus arteriosus, and persistent foramen ovale [38]. Third trimester and term infants with congenital infections were reported to have fever, hyperpigmentation and CNS involvement including meningoencephalitis, microcephaly, seizures, and developmental delays [38]. Genetic studies of these Sri Lankan outbreak strains have indicated that they were of the East/Central/South African (ECSA) lineage with novel mutations that formed a unique cluster, completely separate from other CHIKV isolated in other areas of Sri Lanka and in subsequent outbreaks [39,40]. While this virus lacked the A226V mutation, the virus exhibited unique mutations at nsP1, nsP2, and the capsid protein, which have yet to be characterized [40].

Conversely, the 2005 CHIKV outbreak in Karnataka, India, the 2006–2007 outbreak in Gabon, the 2008–2009 CHIKV outbreak in Thailand, and the 2010 outbreaks in Senegal and India reported dengue-like illness or undifferentiated fever with rash (Table 1) [34,41,44,45]. Furthermore, CHIKV infection of fetuses, neonates, or children caused no significant effects or long-term sequalae [44]. This is puzzling since the ECSA genotype was identified as the cause of all but one of these outbreaks [34,35,41,42,43,48,49]. Furthermore, the Thailand 2008 and the India 2010 outbreak strains were shown to have the same A226V mutation identified in the Reunion Island 2005 outbreak [42,50]. This suggests that the A226V mutation probably doesn’t contribute to viral pathogenesis and that there are likely uncharacterized viral mutations present in these outbreak genomes that are contributing to disease.

## 4. CHIKV in Neonates and Children after Global Expansion

With the awareness of the potentially serious consequences of neonatal and pediatric CHIKV infections, much greater attention is now given to arboviral infection in infants and children. The identification of CHIKV in neonates has occurred more frequently in recent outbreaks and infection typically occurs during delivery (Table 1) [102]. The recent increase in CHIKV infections in children could be explained as a function of better reporting and diagnostics than what was available during the early outbreaks [108,109,110]. However, the scientists who described the early outbreaks were fully capable of documenting febrile disease in pregnant mothers and newborns. They provided comprehensive reports of severe disease and death in the young and old, and they documented abnormal syndromes thus, why would they not have reported symptoms seen in today’s outbreaks if they were present? The modern technology explanation is not sufficient as early scientists used techniques such as ELISA and plaque reduction neutralization tests which are still used today and are gold standards for identifying CHIKV as a causative agent [111,112]. It is far more likely that virus evolution during reemergence and expansion has resulted in increased pathogenesis in man.

Prior to the epidemics in 2005 and the emergence of CHIKV in the Western Hemisphere, there were three distinct lineages of CHIKV that were described [28]. These lineages are anchored to geographical locations where circulating isolates showed little genetic variation [25,28]. ECSA with the A226V mutation is thought to be the most pathogenic, CHIKV-Western Hemisphere (CHIKV-WH) of intermediate pathogenesis, and the unmodified ECSA and West African strains the least pathogenic [113,114]

With the spread of CHIKV into South America and the Caribbean, and the invasion of the ECSA lineage into India, multiple reports of CHIKF in children have been published since 2014. Now, it is relatively common for congenital and pediatric CHIKF to have debilitating, permanent consequences (Table 1).

### 4.1. The West African Lineage

This genotype circulates in the western sub-Saharan regions of Africa and hasn’t been identified outside of this region. It has been documented in periodic outbreaks in Senegal and Nigeria and is typically associated with dengue-like illness accompanied by arthralgia [26,27,45,90,115,116,117,118]. In children, the disease is a minor and self-limiting febrile illness with no debilitating or neurological sequelae reported, even in recent outbreaks (Table 1) [45].

### 4.2. The East Central South African lineage

Historically, this genotype has been identified in all other regions of Africa. It was the virus identified in the early accounts described as a dengue-like illness punctuated by severe arthralgia [4,11,27]. In Africa, the pathology of this genotype hasn’t changed as recent outbreaks report similar syndromes to epidemics during the 1900s (Table 1) [7,41,51,52,59,89,90,91]. However, once this virus was imported into new regions, mutations contributing to its spread were identified. Mutations contributing to increased pathogenesis have not been identified though recurrent genetic motifs have been documented through efforts to track the movement of CHIKV into new regions.

#### 4.2.1. E1: A266V

The 2005 outbreak strains of CHIKV in Reunion Island possessed the E1: A266V mutation [3]. Vector studies showed that this mutation allowed for the virus to be transmitted by *Ae. Albopictus*, which led to a significant expansion in virus geography and emergence in temperate regions [106]. Thus, it was easy to associate this mutation with severe disease, especially when CHIKV with this mutation was isolated from neurologic and severe cases [119]. This assumption was supported by other CHIKV outbreaks with severe disease that also reported the E1 A226V mutation [42,50,54]. However, this mutation has not appropriately been characterized for pathogenesis. Studies aimed at characterizing the A226V mutation in neuroblastoma cells and mice found significant differences in innate immune responses [107,120]. However, these studies used clinically isolated CHIKV-ECSA with several other known mutations that could also contribute to severe disease [107].

#### 4.2.2. E1: K211E

During November 2015 in Brazil, a newborn infected with CHIKV ECSA K211E mutation developed a macular erythematous rash and fever, which progressed into encephalitis with generalized seizures [77,84]. Fortunately, the infant fully recovered after a few weeks and no long-term sequelae were reported [84]. A short distance away in Salvador, Brazil, two vertically infected infants developed a rash, fever, and signs of bacterial infection but recovered within a month after delivery, even though one of the infants had developed a grade 1 intracerebral hemorrhage [83]. In 2016, a new E1: K211E mutation was identified in India, Bangladesh, and Brazil through phylogenetic studies (Table 1) [55,95]. It was shown that the E1: K211E mutation correlated with enhanced viral transmission in *Ae. aegypti* though studies linking this mutation with human disease have not been published [55]. Furthermore, published alignments show that many other mutations are present in these outbreak isolates.

#### 4.2.3. E2: V264A

When CHIKV emerged in the Middle East in 2011, it was shown that the ECSA lineage had lost the A226V mutation and instead encoded a change at E2: V264A, which enhanced viral transmission in *Ae. aegypti* [55]. In India, severe manifestations of the ECSA genotype with the E1: K211E and E2: V264A mutations were reported in congenital and neonatal infections (Table 1) [55]. Specifically, encephalitis, hyperpigmentation, and severe cutaneous lesions were commonly seen in neonates and infants [55,93,94,121,122,123,124]. Other common symptoms included lethargy, poor feeding, and hypotonia [94,123]. In one study, three infants out of 13 had cystic encephalomalacia or diffuse cerebral atrophy three months following discharge [94]. In another report of vertically infected twins, both developed encephalopathy and seizures and MRI scans showed hemorrhagic leukoencephalopathy [125]. Both patients needed support from a ventilator, but both eventually fully recovered [125]. Reports of novel manifestations of CHIKF in infants and toddlers were also made as children began presenting with Stevens–Johnson syndrome and toxic epidermal necrolysis [93]. While neither the K211E nor the V264A are associated with increased human virulence, other CHIKV isolated during this period in this area possessed mutations located at virulence epitopes [55]. While there is likely a correlation, we could not connect these mutations with the isolates described in the above reports.

### 4.3. The Asian Lineage

The Asian genotypes historically circulated India, Indonesia, Singapore, Vietnam, Thailand and other lands in the West Pacific and East Indian Ocean [47]. Prior to the 2005 reemergence of CHIKV, the Asian genotype was commonly misdiagnosed as dengue [12]. Unlike the African genotypes, the Asian lineage was commonly associated with outbreaks of febrile illness punctuated by hemorrhagic fever (Table 1) [5,10,21,23,24,126]. Why this particular lineage produces hemorrhagic manifestations has not been explained, though it is most likely linked to a specific genetic motif present in this linage that has yet to be identified.

### 4.4. The Western Hemisphere Genotype of CHIKV (CHIKV-WH)

The CHIKV-WH genotype is a descendant of the Asian lineage and contains a four amino acid deletion in nsP3, an amino acid insertion at the 3’ UTR, and substitutions at E2: 368 and 6K: 20 [69,127]. In CHIKV, the nsP3 gene encodes a phosphoprotein required for RNA replication and is the major contributor to the virulence of alphaviruses because it interacts with a variety of host proteins (reviewed by Lark et al. [128]). The hypervariable region of nsP3 determines cell specificity for infection and replication [129]. The deletion of the nsP3 codons 379–382 is located within the hypervariable region of the protein [128,130]. Deletions within the hypervariable region of nsP3 in other alphaviruses has resulted in attenuation of virulence in mosquitos and decreased neurovirulence in mice [131,132,133]. In CHIKV, deleting even a single element of nsP3 can reduce replication [129]. It remains unknown how a 4-codon deletion has not affected the virulence of the CHIKV-WH genotype.

#### 4.4.1. Uncharacterized CHIKV-WH

In 2014–2015, CHIKV-WH invaded Mexico and in July 2015 caused an outbreak in the Yucatan region. Here, a one-month old infant was admitted to hospital with CHIKV-WH infection [134]. He presented with fever, lethargy, an erythematous maculopapular rash, and was diagnosed with septic shock secondary to CHIKF [134]. The infant soon developed poor perfusion and was connected to a ventilator [134]. A few hours later, he developed tonic-clonic seizures and fatal septic shock [134]. The island of Curacao experienced an outbreak in 2014 in which three vertically infected neonates experienced severe CNS complications [74]. One infant experienced subdural and intraventricular hemorrhage; the cerebral bleeding was so extensive that the infant died [74]. A second infant developed fever, rash and seizure caused by diffuse white matter lesions [74]. At follow-up one year later, the infant had recovered [74]. A third infant developed fever, rash, and irritability four days after birth and fully recovered after 10 days [74].

#### 4.4.2. E1: K211E

As with the K211E mutation in the ECSA lineage, this mutation also occurs within the CHIKV-WH genotype. This mutation has not been characterized for pathogenesis within this particular genotype though it is thought to be analogous to the K211E mutation in the ECSA lineage, which confers increased infectivity in *Ae. aegypti* [67,135]. Neonates and infants infected with this genotype during the 2014 outbreak in Norte de Santander, Colombia presented with fever, cutaneous eruptions, and diarrhea [64]. Many infants developed severe mucocutaneous, ulcerative lesions, the majority of which occurred in the genital and perianal rejoins [64]. 

#### 4.4.3. E2: L248F

Most CHIKV-WH strains analyzed from the 2013–2015 outbreak in Colombia have possessed both the K221E and the E2: L248F mutations, which are also present in contemporary ECSA isolated from India and the Middle East. There is no phenotype associated with this mutation and it is unknown if this substitution contributes to viral pathogenesis [67]. It is unclear how this substitution would induce a phenotypic change since both amino acids have short non-polar side chains. Nevertheless, CHIKV with this genotype is linked to severe congenital and neonatal CHIKV disease. In Cartagena, Colombia, infants up to 24 months admitted to the emergency department were reported to have febrile illness easily confused with dengue virus [61]. Over the same time period, seven women from Sincelejo, Colombia infected with CHIKV-WH delivered eight babies [62,77]. The mothers presented with mild dengue-like illness, but the newborns presented with maculopapular rash (one with bullous dermatitis) [62]. Severe disease including meningoencephalitis, necrotizing enterocolitis, myocarditis, and respiratory distress were reported for the infants [62]. Three of the eight infants died including two infants with necrotizing enterocolitis [62]. In Segovia, Colombia, a vertically infected infant developed fever, rash and respiratory distress requiring ventilator support [136]. The patient fully recovered with no long-term sequelae reported at the six-month follow-up [136].

#### 4.4.4. E2: A406V

The 2014 Jamaican CHIKV-WH strain caused excessive deaths, especially in children under four years of age, during the 2014 outbreak [70]. The death toll in Jamaica was 10 times greater than the reported CHIKV-fatalities of all Western hemisphere reports combined [70]. Many infants and children presented with fever, arthralgia, and a marked absence of rash or other cutaneous symptoms [68]. A case report of two vertically infected infants described that both developed poor perfusion leading to ischemic fingers and toes [71]. In addition, both infants developed significant idiopathic abdominal distension leading to rectal prolapse [71]. Genetic analysis of this strain showed a valine substitution at residue 406 of the E2 protein [69]. This is a homologous non-polar:non-polar substitution which should have little, if any, effect on the biology of the virus. Further examination of these isolates will likely yield more potential mutations associated with disease.

#### 4.4.5. nsP1: K224N

In 2015, a three-month old infected with CHIKV was admitted to the emergency room after returning from Honduras with fever, rash, respiratory distress and edema [137]. The infant was placed on a ventilator and treated with antibiotics, vasoactives, and fluids with a diagnosis of septic shock secondary to CHIKV infection [137]. The patient fully recovered and was discharged after 10 days [137]. This child was shown to be infected with CHIKV with a K224N substitution of the nsP1 gene. The nsP1 gene in alphaviruses encodes a Rossman-like methyltransferase that is involved in protein capping [130]. nsP1 is also involved in host membrane associations [130,138]. How or if modifications to nsP1 affect CHIKV virulence is unknown, but the substitution of a basic lysine for a polar, amidic asparagine could likely cause significant modifications to the protein structure and function. The K224N mutation is located in a portion of the protein associated with host cell binding [138,139]. Recent work to identify viral resistance to polyamine depletion in mammalian cells identified a mutation G230R, which was found to enhance viral binding with host cells [139]. Perhaps this mutation confers increased infectivity in humans.

#### 4.4.6. nsP3: I285V

The substitution of a non-polar isoleucine for a non-polar valine with a similar side chain would not likely cause any significant changes in protein structure. However, this mutation occurs within the alphavirus unique domain, which contains a zinc coordination site within a protein fold [140]. Examinations of this domain have indicated that it is necessary for viral replication and mutations in this region are unstable and reduce virulence [140]. This could be a cause of the reduced pathogenicity reported in Puerto Rico, 2014. Here, perinatal transmission of this particular CHIKV-WH genotype resulted in rash followed by eczema [72].

#### 4.4.7. nsP4: R99Q

During the 2014–2015 outbreak in Guiana, 26 febrile children under three months of age infected with CHIKV-WH with a R99Q substitution at nsP4 were hospitalized [76]. The most common symptoms reported were fever, rash and edema of the hands and feet indicating possible problems with perfusion [76]. The nsP4 gene is the most highly conserved gene of the alphaviruses and encodes an RNA-dependent-RNA-polymerase that functions in producing the synthetic properties of the viral replicase complex [130,141]. Research investigating amino acid substitution of this protein has shown reduction in viral replication [141]. The R99Q substitution replaces a positively charged, basic arginine with an electronegative, polar glutamine. This change could cause changes to protein structure and function, though it is unknown how this mutation may have contributed to disease.

#### 4.4.8. nsP4: A459V

This substitution was associated with CHIKV-WH circulating in Haiti during 2014-2015 [69]. In a report describing arboviral infections in school-age children, the major symptoms seen were fever, arthralgia, and myalgia [73]. Rash was observed in only 4% of patients [73]. As with other substitutions discussed, the alanine to valine substitution reflect non-polar aliphatic to non-polar aliphatic and should not cause any major changes to the protein structure.

## 5. Consequences of Co-Infection

Case reports have indicated that congenital co-infection of CHIKV with another pathogen significantly increases the pathology of infection and, in most cases, leads to miscarriage. However, co-infections in children do not show increased pathology. Following the 2015 CHIKV and Zika outbreaks in Colombia, a 25-week pregnant mother was diagnosed with *Toxoplasma gondii*, Zika virus, and CHIKV-WH in her amniotic fluid [142]. Severe neurological and physical deformities were identified resulting in early termination of pregnancy [142]. In March the following year, a female in her second trimester was diagnosed by RT-PCR with a CHIKV-WH-Zika virus co-infection following an abnormal sonogram, which indicated no fetal heartbeat [143]. An autopsy of the fetus showed low weight along with renal and placental calcifications [143]. Conversely, a three-year old patient from New Delhi was co-infected with CHIKV ECSA, dengue, and *Plasmodium vivax* but fortunately was not ill enough to warrant hospitalization [144]. The strain circulating in the area at the time of this child’s infection contained the E1: K211E and E2: V264A mutations, which were identified in pediatric and neonatal cases with severe cutaneous manifestations. Furthermore, infants infected with both CHIKV-WH and dengue virus during the 2014 outbreak in Colombia did not exhibit more serious manifestations of disease [64]. In Haiti as well, school-age children co-infected with CHIKV-WH and Zika virus did not exhibit worse disease than singly infected patients [73].

## 6. Conclusions

CHIKV has been reported as a cause of febrile illness in children and neonates since the 1960s. Following the reemergence of CHIKV in the regions surrounding the Indian Ocean and the expansion of CHIKV into the Western Hemisphere, CHIKV-WH infections in children and neonates are reported more frequently and with more severe syndromes and sequelae (Table 1). Furthermore, congenital CHIKV infection has evolved from causing minor, self-limiting disease to resulting in multiple severe manifestations and death (Table 1). While severe CHIKV disease can be linked to unique viral genotypes, the contribution of novel mutations to disease pathogenesis has yet to be described. For instance, the K221E mutation occurs in both the ECSA and CHIKV-WH genotypes and has been identified in neonatal and pediatric infections with severe cutaneous lesions (Table 1). Furthermore, if the K211E mutation, is combined with a second mutation (ECSA E2: V264A and CHIKV-WH E2: L284F), there is a marked increase in disease severity, though it is unclear if these mutations are the cause (Table 1). Characterization of other mutations present in these isolated genomes has not been performed. This is unfortunate as the non-structural proteins have been shown to be involved in viral pathogenesis and neuroinvasiveness more frequently than envelope proteins due to their myriad of functions in viral replication, whereas the functions of the structural proteins are for host cell recognition, binding, and entry [131,132,133,138,139,140,145,146,147,148,149]

This review highlights the limitations of our knowledge regarding CHIKV pathogenesis. This is likely a result of the disconnect between clinical and basic science. In clinical research, the main focus is diagnosis, treatment, and prevention. Viral disease is often diagnosed via serological or nucleic acid tests that focus on small portions of the genome (usually the envelope protein). Viral lineages and mutations are rarely emphasized unless the knowledge could direct patient care or impact diagnostic assays. Point mutations are usually ignored as long as a definitive diagnosis can be made. On the contrary, basic phylogenetic research focuses primarily on the genetic sequence. Often times, genomes are derived and characterized from insect samples and, when obtained from patient specimens, little if any attention is given to the clinical profiles of patients. Furthermore, very few viral genomes have been obtained from children and neonates given their inherent vulnerability as a research population.

It is of interest that divergent and geographically distinct CHIKV genotypes can induce similar disease syndromes. If a single mutation can confer increased vector competence, as with the A226V mutation, then surely there are viral mutations that contribute to viral replication in skin, placental, or CNS tissues. There are hundreds, if not thousands, of mutations that have yet to be characterized. Through the use of site-directed mutagenesis of infectious clones or sub-genomic replicon strategies, it is possible to evaluate functional mutations in a silent background. With over 6100 CHIKV sequences available on GenBank, we just need to look.

## Figures and Tables

**Figure 1 viruses-11-00294-f001:**
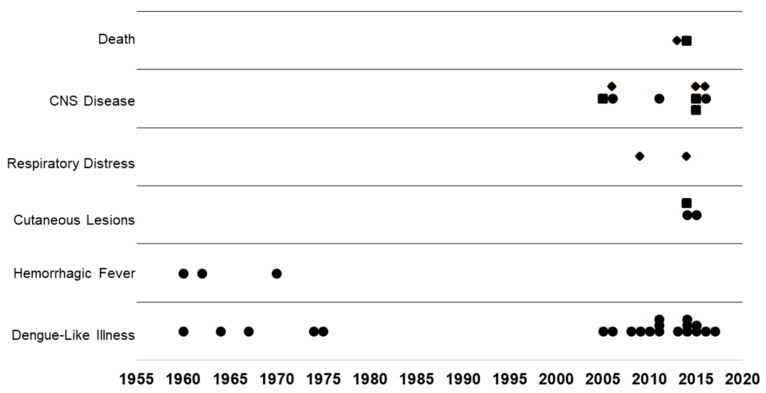
Consequences of congenital, perinatal, and pediatric chikungunya infections are increasing. Chikungunya infection in newborns, infants, and children documented since 1960 are plotted against the most severe symptoms or sequelae identified in the associated report.

**Table 1 viruses-11-00294-t001:** Symptoms of chikungunya fever are associated with new viral genotypes. Chikungunya outbreaks with descriptions of chikungunya fever in children, infants, and neonates are matched with the genotype circulating in the region at the time the outbreak was reported. The contribution of these mutations to disease has not been explored.

Year	Location	Disease Presentation	Viral Lineage	Citations
1960	Philippines and Thailand	Hemorrhagic fever	Asian	[5,25]
1960	Nigeria	Dengue-like illness	West African	[25,26,31]
1962–1964	Thailand	Dengue-like illness with hemorrhagic fever	Asian	[20,21,22,23,24]
1964	India	Arthralgia, rash, fever	Asian	[17,19,25]
1967	Vietnam	Dengue-like illness	Asian	[18,25]
1970	Burma	Hemorrhagic fever	Asian	[9,25]
1974	Nigeria	Dengue-like illness	West African	[25,26]
1975	South Africa	Dengue-like illness	ECSA	[25,32,33]
2005	India	Arthralgia, rash, fever	ECSA	[34,35]
2005	Reunion Island	Perinatal infection Rash > 3 CNS < 3	ECSA E1: A226V	[3,36]
2006	Bellary, India	CNS infections	ECSA E1: K211E E1: A226V	[37]
2006/2007	Sri Lanka	Perinatal and Congenital infections Pre-term birth Hyperpigmentation, cardiac defects, CNS symptoms and defects	ECSA NSP1: R488Q NSP2: A331V C: V27I	[25,38,39,40]
2006/2007	Gabon	Dengue-like illness	ECSA	[41]
2008	Thailand	Dengue-like illness	ECSA E1: A226V	[42,43,44]
2009/2010	Senegal	Dengue-like illness	West African	[25,45,46,47]
2009/2010	India	Dengue-like illness 1 case report of vertical transmission resulting in respiratory distress, fever, and maculopapular rash	ECSA E1: A226V	[48,49,50]
2010	Gabon	Rash	ECSA	[51,52]
2011	Cambodia	Dengue-like illness sporadic reports of Acute Meningoencephalitis	ECSA E1: A226V	[53,54]
2011	Yemen	Fever, arthralgia, rash	ECSA E2: V264A	[55,56]
2011	Pakistan	Fever, arthralgia, rash	ECSA E2: V264A	[55,57]
2011	Tanzania	Dengue-like illness	ECSA	[58,59]
2013/2014	Tanzania	Fever, rash, cough	ECSA	[59,60]
2013/2014	Colombia (Cartagena)	Congenital infections respiratory distress, necrotizing enterocolitis, meningoencephalitis, myocarditis, pericarditis and death	Asian-WH	[61,62]
2014	Mozambique	Dengue-like illness	ECSA	[63]
2014	North Eastern Colombia	Severe mucocutaneous lesions	Asian-WH E1: K211E	[64,65,66,67]
2014	Jamaica	Fever, arthralgia, excessive deaths Vertical transmission	Asian-WH E2: A406V	[68,69,70,71]
2014	Puerto Rico	Perinatal Transmission—Eczema, Fever	Asian-WH NSP3 I285V	[69,72]
2014	Haiti	Dengue-like illness	Asian-WH NSP4 A459V	[69,73]
2014	Curacao	Neonatal seizures intracerebral bleeding, death	Asian-WH	[74,75]
2014/2015	French Guiana	Fever, Rash, edema of the extremities	Asian-WH NSP4: R99Q	[69,76]
2014/2015	Nicaragua	Rash arthralgia	Asian-WH	[77,78,79]
2014/2015	Ecuador	Rash arthralgia	Asian-WH	[69,80]
2015	Colombia	Infant CNS, Rash > 3	Asian-WH	[77,81]
2014/2015	Brazil	100% children with exanthema and open blisters. Neonates with fever, lethargy, pulmonary edema, encephalitis	Asian-WH	[82,83,84,85,86]
2014/2015	Honduras	Fever, Rash, seizures, meningoencephalitis	Asian-WH NSP1: K224N	[69,87]
2015	Yucatan	Fever, Exanthema, blisters	Asian-WH	[82,88]
2015/2016	Mozambique	Dengue-like illness	ECSA	[89]
2015/2016	Nigeria	Dengue-like illness	West African	[90]
2016	Kenya	Dengue-like illness	ECSA	[91]
2016	India	Fever, Rash (erythematous maculopapular, purpuric, pustular, toxic epidermal necrolysis) Stevens–Johnson syndrome Hyperpigmentation Neonatal seizures	ECSA E1: K211E and E2: V264A	[55,92,93,94]
2016	Brazil	Dengue-like illness. 1 case report of vertically transmitted neonatal encephalitis.	ECSA E1: K211E and E2: V264A	[69,84,95,96]
2017	Bangladesh	Dengue-like illness	ECSA E1: K211E and E2: V264A	[55,97]

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
