# Peer review of "Chikungunya in Infants and Children: Is Pathogenesis Increasing?"

_viruses, 2019, doi:10.3390/v11030294_

Reviewer 1 Report

Barr and Vaidhyanathan have provided a concise overview of chikungunya virus (CHIKV) infections in infants and children and highlight potential virus lineage-specific determinants related to pathogenicity and virulence.

Overall, this is an interesting review of the current state of knowledge regarding CHIKV infections in infants and children. The authors have presented a thorough overview of the current literature and the relation of viral lineage to clinical manifestations of disease.

There are several moderate/minor concerns that should be addressed by the authors prior to acceptance of the manuscript. These will be detailed below:

CHIKV vs chikungunya fever: the authors should ensure that they properly address whether they are speaking of the virus or the illness in the different sections of the manuscript. CHIKV infections result in Chikungunya fever (CHIKF); CHIKV is not itself the illness.

Amino acid descriptions: the authors should use the standard 3 letter codes when describing the amino acids (outside of the single letter descriptors e.g. A226V).

Numeral inconsistencies: there were many instances where the authors switched between presenting numerals vs spelling out the number. With the exception of the start of a sentence it may be easiest for the authors to use numerals throughout. For examples, 4 vs four in lines 205 vs 214.

Genus and species presentation inconsistencies: the authors should note that after the initial introduction of Aedes spp. in lines 25-26, they should use the shorthand versions from that point onwards (i.e. Aedes albopictus is continually presented in full after this point).

Paragraph/section presentations: some inconsistencies that should be corrected. For example, line 172 vs 177. Italics missing in the header for line 216.

Additional minor comments and clarifications:

avoid the use of contractions (e.g. hasn't)

line 74-75: when speaking of hemorrhagic manifestations of influenza viruses, the authors should specifically state that they are speaking of pulmonary hemorrhage (assuming this is what they are referring to).

line 148: "...early scientists use techniques...". Such as what techniques? 

line 174: italicize Ae. albopictus 

line 186: capitalize Middle East

line 197: "presenting to dermatologists" does not need to be there. 

line 204: replace descendant with "is a descendant of" or "descended from"

line 205: "Non-structural" does not need to be capitalized. NSP should also follow this in parentheses.

line 207: the protein encodes a phosphoprotein? Should this not be "the gene encodes a phosphoprotein"?

line 232: replace Aedes with Ae. 

line 237: this is the first introduction of the E2: L248F mutation. This should be introduced in more detail here. 

line 242-243: Dengue virus is not the illness. 

lines 258-260: based on this logic should the A-to-V mutation in E1 have any effect on the biology of the virus?

line 267: the authors list "vasoactive" as a treatment. Do they mean vasoactivators?

lines 296 and 302: Zika virus should be stated, not just Zika.

line 306: the authors say "CHIKV in the Indian Ocean...". Do they mean in the regions surrounding the Indian Ocean?

line 314: replace CHICK with CHIKV

Author Response

the authors should ensure that they properly address whether they are speaking of the virus or the illness in the different sections of the manuscript. CHIKV infections result in Chikungunya fever (CHIKF); CHIKV is not itself the illness.

We have made edits though out the text replacing CHIKV with chikungunya fever (CHIKF) when referring to illness caused by CHIKV.

Amino acid descriptions: the authors should use the standard 3 letter codes when describing the amino acids (outside of the single letter descriptors e.g. A226V).

We have made the requested changes, though we used the full amino acid name instead of the 3 letter code to make it easier for readers.

Numeral inconsistencies: there were many instances where the authors switched between presenting numerals vs spelling out the number. With the exception of the start of a sentence it may be easiest for the authors to use numerals throughout. For examples, 4 vs four in lines 205 vs 214.

We have made the requested changes.

Genus and species presentation inconsistencies: the authors should note that after the initial introduction of Aedes spp. in lines 25-26, they should use the shorthand versions from that point onwards (i.e. Aedes albopictus is continually presented in full after this point).

We have made the requested changes.

Paragraph/section presentations: some inconsistencies that should be corrected. For example, line 172 vs 177. Italics missing in the header for line 216.

We have made the requested changes.

line 74-75: when speaking of hemorrhagic manifestations of influenza viruses, the authors should specifically state that they are speaking of pulmonaryy hemorrhage (assuming this is what they are referring to).

added pulmonary before hemorrhage

line 148: "...early scientists use techniques...". Such as what techniques?

Added ELISA and plaque reduction neutralization tests  

line 174: italicize Ae. albopictus 

Done

line 186: capitalize Middle East

Done

line 197: "presenting to dermatologists" does not need to be there. 

Deleted

line 204: replace descendant with "is a descendant of" or "descended from"

Done

line 205: "Non-structural" does not need to be capitalized. NSP should also follow this in parentheses.

Done

line 207: the protein encodes a phosphoprotein? Should this not be "the gene encodes a phosphoprotein"?

Fixed

line 232: replace Aedes with Ae. 

Done

line 237: this is the first introduction of the E2: L248F mutation. This should be introduced in more detail here. 

Done

line 242-243: Dengue virus is not the illness. 

Done

lines 258-260: based on this logic should the A-to-V mutation in E1 have any effect on the biology of the virus?

Explained at the end of paragraph.

line 267: the authors list "vasoactive" as a treatment. Do they mean vasoactivators?

A vasoactive substance is an endogenous agent or pharmaceutical drug that has the effect of either increasing or decreasing blood pressure and/or heart rate through its vasoactivity, that is, vascular activity (effect on blood vessels).

lines 296 and 302: Zika virus should be stated, not just Zika.

Done

line 306: the authors say "CHIKV in the Indian Ocean...". Do they mean in the regions surrounding the Indian Ocean?

Fixed

line 314: replace CHICK with CHIKV

Done

Reviewer 2 Report

Following the epidemic in Reunion Island, chikungunya virus (CHIKV), CHIKV distribution areas have rapidly expanded and the number of cases have increased. A single amino acid substitution in E1 is proved to enhance transmission by Ae. albopictus mosquito. Along with the expansion, the number of severe cases, which exhibit CNS manifestation in infants and neonates, has increased. Authors reviewed the infant and neonate cases, and discussed relationship of pathogenicity and particular genotype. The prepared manuscript needs to be revised.

 Major comments

1.       Please describe CHIKV genotype/lineage and genome organization (brief description of each viral proteins) in the Introduction or put these sections prior to case reviewed sections.

2.       5. Consequences of co-infections section (lines 288-303): Were any mutations in the CHIKV genome identified in the co-infection cases? If any, please described here.

3.       In the Table1, each case is arranged in year order when the case was reported. If author can rearrange this table to categorize by the lineage first, then aligned in year order, it may give better understanding the relationship of lineage, mutation and pathogenicity.

4.       Among mutations authors reviewed in the manuscript, mutations in E1 and E2 are more likely to link with severe manifestations. Do authors agree? If so, please speculate more on this.

5.       Is there any reported mutation that provides better replication in placenta, umbilical cord etc that may be relate to vertical transmission?

6.       In conclusion, author had better providing future direction to understand genotype-pathogenicity relationship. For example, CHIKV infectious clone/subgenomic replicon technologies can be applied in this field.

 Minor comments

1.       “Dengue” should be described as “dengue”.

2.       Line 65 and 80: 104C should be 104F.

Author Response

       Please describe CHIKV genotype/lineage and genome organization (brief description of each viral proteins) in the Introduction or put these sections prior to case reviewed sections.

Information regarding CHIKV lineage has been added to the introduction and expanded upon in section 4.

2.       5. Consequences of co-infections section (lines 288-303): Were any mutations in the CHIKV genome identified in the co-infection cases? If any, please described here.

descriptions for associated viruses were added though there were no mutations reported.

3.       In the Table1, each case is arranged in year order when the case was reported. If author can rearrange this table to categorize by the lineage first, then aligned in year order, it may give better understanding the relationship of lineage, mutation and pathogenicity.

I tried this and it made the information more confusing.  I left it in chronological format so that the reader could easily find the information.  I did however add more discussion to the text regarding identical mutations in different lineages in sections 4.4.2 and 4.4.3.

4.       Among mutations authors reviewed in the manuscript, mutations in E1 and E2 are more likely to link with severe manifestations. Do authors agree? If so, please speculate more on this.

We added text throughout the manuscript explaining that the mutations linked with these outbreaks were not likely contributing to disease. We explain that other mutations are present in these isolates but have nit been characterized.  We also explain it as a function of the disconnect between clinical and bioinformatics research.

5.       Is there any reported mutation that provides better replication in placenta, umbilical cord etc that may be relate to vertical transmission?

we could not find any research regarding this but we are currently working on this in our lab.  Currently, all work is on Zika

6.       In conclusion, author had better providing future direction to understand genotype-pathogenicity relationship. For example, CHIKV infectious clone/subgenomic replicon technologies can be applied in this field.

We added text to the conclusion that there are lots of mutations needing characterization and that it should be done in a silent background using an infectious clone or sub-genomic replicons.

 Minor comments

       “Dengue” should be described as “dengue”.

Done

2.       Line 65 and 80: 104C should be 104F.

Done

 Reviewer 3 Report

In the review by Barr et. al. the authors discuss the incidents of CHIKV infection in infants and children.  Overall, the subject matter is important, relevant and therefore should be published.  However the manuscript as is requires revision.  The overall organization needs to be improved.  The authors attempted to combine viral genetics, outbreak information and disease pathology but there is clear lack of cohesiveness to the text.  This needs to be fixed.  The authors offer little in way of perspective when describing the disease presentation, viral lineage and mutations.  This topics were never really connected or expanded upon.  For example, no perspective, insight or hypothesis was given in regards to the hemorrhagic fever observed in the Asian lineage and no where else.  What do they believe is happening there? 

itemized concerns:

-The authors were not consistent in numbers nomenclature- going back and forth between 1 and One through out the text.

-While the authors clearly define congenital and perinatal infection in the introduction, figure 1 includes pediatric infection, which was not defined.

-Figure 1 needs to be re-done as there appear to be symbols behind the black symbols.  Also the way the years are separated might be made clearer with lines to section them off.

-Line 66- Typically if one starts a sentence with a number it is spelled out.

-Terms like Toll-like receptor (Line 100), non-structural protein 1 (Line 126) are typically accompanied by there common abbreviations, TLR3 or nsP1

-Line 206- NSP-3 is written however later in Lines 209 and 210 NSP3 is written. The protein is typically denoted nsP3 in the literature

-All of the italicized sub-headings are transitioned into awkwardly. Simply list the term, then begin the sentence on its own.

-Why does E1: K211E have two separate sections? (Lines 177-185), then again on Lines 228-235?

-Line 240- Linked in missing an L.

-The sub-sections that discuss the individual nsPs is fairly superficial and should be enhanced, for example can insight be gained from available structural information on other alphaviruses or proteins with similar functional properties?

Author Response

The authors offer little in way of perspective when describing the disease presentation, viral lineage and mutations.  This topics were never really connected or expanded upon.  For example, no perspective, insight or hypothesis was given in regards to the hemorrhagic fever observed in the Asian lineage and no where else.  What do they believe is happening there? 

We added text throughout the draft indicating that there is a lack of research in this area and that there are likely mutations associated with these various lineages.

-The authors were not consistent in numbers nomenclature- going back and forth between 1 and One through out the text.

Fixed

-While the authors clearly define congenital and perinatal infection in the introduction, figure 1 includes pediatric infection, which was not defined.

Defined

-Figure 1 needs to be re-done as there appear to be symbols behind the black symbols.  Also the way the years are separated might be made clearer with lines to section them off.

-Line 66- Typically if one starts a sentence with a number it is spelled out.

Fixed throughout text.

-Terms like Toll-like receptor (Line 100), non-structural protein 1 (Line 126) are typically accompanied by there common abbreviations, TLR3 or nsP1

Fixed throughout text.

-Line 206- NSP-3 is written however later in Lines 209 and 210 NSP3 is written. The protein is typically denoted nsP3 in the literature

Fixed throughout text.

-All of the italicized sub-headings are transitioned into awkwardly. Simply list the term, then begin the sentence on its own.

Fixed throughout text.

-Why does E1: K211E have two separate sections? (Lines 177-185), then again on Lines 228-235?

discussion regarding this is added to the text as we find it of interest that 2 genetically distinct CHIKV lineages acquire the same mutation following global expansion.

-Line 240- Linked in missing an L.

fixed

-The sub-sections that discuss the individual nsPs is fairly superficial and should be enhanced, for example can insight be gained from available structural information on other alphaviruses or proteins with similar functional properties?

Additional information was added to the description of non-structural proteins.  There wasn't much we felt comfortable adding as genetic studies have shown CHIKV to be isolated from other alphaviruses.  However we did add text regarding the pathogenesis of Semliki forest virus for with there is lots of information.  We chose Semliki Forest because it is also an old world alphavirus.